# Uncertainty and Tourism Consumption Preferences: Evidence from the Representative Chinese City of Shenzhen

**Xuemin Liu [1], Jiaoju Ge [2],\*  and Ting Ren [1]**

1   HSBC Business School, Peking University, Shenzhen 518055, China; liuxm@phbs.pku.edu.cn (X.L.); renting@phbs.pku.edu.cn (T.R.)
2   School of Economics and Management, Harbin Institute of Technology Shenzhen, Shenzhen 518055, China
\*   Correspondence: jiaoge@hit.edu.cn

**Abstract:** As the fastest growing city in China, Shenzhen, a pioneer of reform and opening up, seems to have exhausted its advantages of geographical convenience, cheap land, and cheap labor; thus, the tourism industry has been adopted as a new economic strategy. However, the Shenzhen tourism market has received little attention from scholars and few studies have been done to investigate the relationships between uncertainty and tourism consumption preferences in different cultures. Therefore, this paper attempts to study Shenzhen as a travel destination to explore the consumption preferences of domestic and international tourists, specify uncertainties in tourism consumption activities, and examine their impacts on preferences. The above aims are achieved using a survey method and a new theoretically proposed preference uncertainty model inspired by combining the modified virtual expert preference approach and the ordered probit model (MVEP-OPM), in which three major components of tourism consumption (food, accommodation, and shopping) are estimated. (1) The results show that there are significant differences in tourism consumption preferences between domestic tourists and international visitors. (2) Inexperience, unfamiliarity, imperfect knowledge, and policy uncertainty are major uncertainties in tourism consumption activities, which affect tourist preferences toward consumption behavior. (3) Uncertainty plays a different moderating role in product-related influential factors, such as sanitation, safety, and decoration.

**Keywords:** uncertainty; tourism consumption preference; Shenzhen

## 1. Introduction

Tourism is becoming a promising industry due to its diverse potential, playing a decisive role in adjusting industrial structure, driving local consumption, stimulating economic growth, increasing revenue, and providing employment opportunities during the process of urban transformation and upgrading [1]. Therefore, the characteristics [2], attitudes [3], behaviors [4], preferences [5], perceptions [6], motivations [7], and satisfaction [8] in tourism consumption have become focuses of attention. Preferences and motivations sometimes overlap, but motivations play a guiding role rather than a decisive role in consumer satisfaction, while preferences refer to a clear attitude towards making choices after detailed comparisons of various products with similar attributes and functions [9], usually accompanied by uncertainties. In other words, uncertainties exist in tourism consumption preferences [10]; for example, when a tourist books accommodation, they are buying an intangible service for which an evaluation has to be made after it has been experienced. Consequently, various methods and calibration techniques have been proposed to measure uncertainty, including: (1) the numerical certainty scale (NCS), using a follow-up question to elicit the degree of respondent uncertainty on a probabilistic scale ranging from 0 to 100 percent [11] or on an ordinal scale ranging from 1 (very uncertain) to 10 (very certain) [12]; (2) the polychotomous choice (PC) approach, introducing a post-evaluation question by providing response categories, such as "definitely yes", "probably yes", "maybe yes", "maybe no", "probably no" and "definitely no", reflecting respondents'

certainty levels [13]; (3) the multiple bound uncertainty [14] and two-way payment ladder [15] methods are also prevalent. Even so, the impact of uncertainty is still difficult to estimate due to the intangibility, inseparability, and heterogeneity of tourism products [16], resulting in growing concerns related to tourism uncertainty tolerance [17], perception [18], and avoidance [19]. Therefore, it is necessary to explore the impacts of uncertainty on tourism consumption preferences.

Shenzhen was established as the first Special Economic Zone in 1979, and emerged as a modern metropolitan area from a tiny border village, named the "Shenzhen miracle". Moreover, Shenzhen was the first city in socialist China to experience the operation of a market economy, placing it far ahead of other regions. In 2018, Shenzhen's GDP reached 2.4 trillion RMB with a growth rate of 7.6%, surpassing Hong Kong and ranking among the top five Asian cities. However, Shenzhen as a "reformer" seems to have exhausted its advantages of privileged policies, geographical convenience, foreign capital, and cheap land and labor, leading to the relocation of giant companies such as Huawei, Zhongxing Telecom Equipment, and Foxconn. Fortunately, the prominent tourism industry promoted by the Guangdong–Hong Kong–Macao Greater Bay Area strategy provides Shenzhen an opportunity to boost its economy, since it has the potential to develop into an internationally renowned tourism city with top-notch influence in terms of image, economy, innovation, and livability. As a "hub city" in the Asia–Pacific region, Shenzhen is the only city in mainland China that possesses sea, land, air, and railroad ports of entry, providing an ideal portal to international inbound tourists. According to the Shenzhen Statistical Yearbook in 2018, both its numbers of overnight domestic (48.15 million) and international (12.07 million) tourists ranked first in China, indicating that numerous tourists choose Shenzhen as a transit route for outbound or inbound travels. However, this huge passenger flow has resulted in a large-scale outflow of tourism expenditure potential rather than stimulating local tourism consumption, accompanied by a continuously expanding deficit in the balance of tourist spending in Shenzhen. Obviously, the lack of cultural heritage, clear positioning, and distinctive attributes have affected Shenzhen's ability to trigger tourism consumption. Taking a look at its competitors, Hong Kong is known as the "shopping paradise", Macau is famous for its gambling and entertainment, and Guangzhou enjoys a great reputation for local cuisine. Therefore, Shenzhen urgently needs to clarify and highlight its own peculiar enchantment as an emerging tourism city through in-depth understanding of tourism consumption preferences.

In summary, with Shenzhen as a travel destination, exploration of the domestic and international tourist preferences as they relate to uncertainty is advantageous and unprecedented. The specific research objectives are: (1) to explore and identify uncertainties and drivers in tourism consumption preferences; (2) to examine differences in preferences between domestic and international visitors; (3) to investigate the impacts of uncertainty on tourism consumption preferences and its moderating role on other influential factors.

The main contributions of this study are firstly that the uncertainty perspective is introduced to the field of tourism consumption and measured with a new proposed theoretical model, inspired by combining the modified virtual expert preference approach and the ordered probit model (MVEP-OPM). To the best of our knowledge, despite the prevalence of uncertainty in tourism consumption preferences [20], it has long been ignored, especially regarding inexperienced visitors with imperfect knowledge [21]. In addition, our proposed MVEP-OPM overcomes the limitations of existing NCS and PC estimation methods, such as subjectivity, low or high raters, and higher rates of "yes" responses. Secondly, it is the first attempt to explore the consumption preferences of domestic and international tourists in terms of food, accommodation, and shopping in a representative Chinese city—Shenzhen. Compared with the increasing literature on Chinese tourists' consumption overseas, little research has touched on international visitors and inbound tourism. Moreover, existing studies tend to focus on developed countries or famous destinations, such as the Maldives, Paris, Hawaii, and South Korea, while ignore emerging forces in the tourism market in mainland China. In fact, Shenzhen possesses beautiful

coastlines, charming beaches, a pleasant climate, a high-tech image, civilized behavior, and metropolitan style. As early as 2010, Shenzhen was rated as one of "31 destinations for global tourists" by the New York Times. In 2019, Shenzhen ranked second in the list of the top 10 best travel cities published by Lonely Planet. Despite frequent appearances on global travel lists, few scholars have focused on the Shenzhen tourism market.

## 2. Literature Review and Conceptual Framework

### 2.1. Tourism Consumption Preferences

According to the 2018 China Statistical Yearbook, food ($10.31 billion, 8.4%), accommodation ($12.21 billion, 9.9%), and shopping ($22.99 billion, 18.6%) are major tourism consumption expenditures for international visitors. Therefore, research on tourism consumption preferences relating to uncertainty will focus on food, accommodation, and shopping.

#### 2.1.1. Food

Local food provides social and leisure attributes to tourism and cultural activities, which are driving forces for tourism consumption and are often described by preference, liking, and choice. Food preference refers to the most satisfying choice within the option range, while food liking is the cause of preference and food choice is a way to express preference [22]. Due to the importance of food preference, its influencing factors have been extensively explored and summarized at the culture level in terms of individual preference, food-related, environment, and experience components [23,24]. The impacts of culture and religion on consumer food preferences are deeply rooted; for example, Muslims are forbidden from eating pork and Western food culture differs from Chinese dining etiquette. For individuals, sociodemographic factors such as age, gender, marital status, and household income are the basis for the diversity of preferences [25]. Flavor, variety, luster, sensory appeal, and other attributes of food can stimulate and delight visitors' senses of smell, taste, and sight [26]. The dining environment is closely related to internal conditions (service level, price, hygiene, and atmosphere) [27] and the external atmosphere (location, evaluation, and recommendations). Regarding the experience, this is complex due to subjective judgments involving health, availability, and convenience, as well as objective factors related to food attributes [23]. Based on the above explanation, taste, variety, service quality, price, location, hygiene, and evaluation are selected as drivers of food tourism consumption preference in this study.

#### 2.1.2. Accommodation

Accommodation is considered to be the main requirements for luxury, leisure, health, and business travel; for example, American tourists spend more than 30% of their total budget on accommodation [28]. Since hotel reservations take precedence over other activities, the selection criteria have been largely explored. For example, Prayag and Lee [7] examined the moderating effects of service interactions, nationality, and hotel star rating on the relationship between tourist motivation and place attachment, while Lai and Hitchcock [29] compared the service quality attributes for stand-alone and resort-based luxury hotels in Macau by surveying 299 overnight visitors. Furthermore, Sohrabi et al. [30] identified the key consumption drivers for hotels in Tehran, including security and protection, promenades, staff and their services, cleanliness and room comfort, networks, pleasure, room facilities, car parking, and expenditure. Additionally, Xu et al. [31] outlined the dimensions of housing, location, host–guest interaction, facilities, and overall sensations as contributing to accommodation sharing through a data-driven approach. One study Hsieh et al. [32] suggested that the distance to commercial districts and sightseeing locations was the most valued aspect by visitors when considering accommodation, while another study Chu and Choi [33] emphasized that both business and leisure travelers showed the same attitudes towards value, room and front desk, food, and recreation components. Furthermore, Rivers [34] highlighted that long-stay guests preferred hotels with memberships and promotions. In addition, Kim et al. [35] suggested that online marketing

efforts should highlight a hotel's identity and accessibility. In view of the above, regarding accommodation tourism consumption, widely used factors are identified in this research, including safety, service, sanitation, price, location, and free breakfasts.

### 2.1.3. Shopping

Shopping can bring about memorable experiences for tourists due to providing relaxation and pleasure, especially for East Asian tourists, who were found to regard shopping as a pleasurable activity that enriches lives [36]. Duty-free stores and famous shopping destinations such as Hong Kong and Hawaii are preferred and attract crazy purchases. Factors influencing shopping have been constantly studied, for example women prefer buying clothes, shoes, jewelry, and cosmetics, while male visitors prefer tobacco, cigarettes, and wine [37]. Married tourists tend to spend more on shopping [2], but low-income groups usually consider their actual needs [38] and are attracted by lower prices. In addition, brands with a long history and reputation are more attractive, despite the higher prices [39]. Product quality provides users with functional or practical benefits, but the service level contributes to consumer satisfaction and commemorative significance contributes to joyful memories [40]. Furthermore, the in-store environment can guide tourists' shopping behavior. For example, color affects perceptions of products [41] and a cluttered environment brings negative emotions to consumers [42]. Store popularity, well-known shopping regions, thematic shopping, and celebrated shopping malls all increase shopping expenses [43]. In addition, promotions largely determine the attitudes of consumers [44]. Therefore, regarding shopping, brand, price, service, quality, packaging, memorable experiences, decorations, practical needs, store popularity, and promotions are selected herein due to their representative nature.

### 2.2. Uncertainty in Tourism Consumption

Uncertainty associated with travel is highlighted as reflecting tourists' concerns and consumption habits, especially for international tourists, for whom uncertainty issues are often amplified [20]. A previous study Quintal et al. [45] discussed the effects of uncertainty on tourism decisions and found that increased uncertainty in Australian visitors negatively affected their motivation to take vacations. To reduce uncertainty, official websites and well-known travel agencies are generally relied on rather than individuals when obtaining information about the travel destination, and tourists tend to travel in groups, picking shorter stays and fewer visits [19]. However, there is still no consensus on how to measure uncertainty. Regarding the NCS, the theoretical support is insufficient in terms of determining the appropriate cutoff points, because numerical probabilities and the levels of an ordinal scale can be interpreted differently by diverse individuals. As for the PC approach, it can be influenced by the words used to elicit respondent uncertainty [15], which might generate higher rates of "yes" responses. Therefore, a new method is proposed, whereby uncertainty is considered a latent variable defined by a set of influential factors and integrated into the virtual expert model [46] to replace the role of virtual experts. Before implementation, it is necessary to explore sources of uncertainty and their impacts on tourism consumption preferences.

### 2.2.1. Inexperience

Insufficient experience and infrequent purchases inevitably increase the level of uncertainty in tourism consumption [21]. Generally, travel experience has a positive effect on consumer well-being and loyalty, which is conducive to reducing the intertemporal instability of uncertainty preferences [47]. Two criteria were available to measure tourism experiences and familiarity in this study, namely the number of trips and the duration of the stay [48]. These two indicators reflect the degree to which a person utilizes opportunities to accrue experience through time, events, and activities, providing an accurate means to distinguish individuals with different levels of travel experience.

### 2.2.2. Imperfect Knowledge

Uncertainty entails a limitation of knowledge on the tourism destination and future conditions, making the outcome of an action impossible to predict. Imperfect knowledge partly stems from the cultural differences, daily habits, and lifestyles of tourists [49]. In other words, familiarity with diverse cultures and destinations is informed by tacit or personal knowledge, which is particularly true for tourists visiting unfamiliar places [20]. Consequently, imperfect knowledge may lead to relatively high uncertainty; the higher level of education, the lower the degree of knowledge imperfection.

### 2.2.3. Policy Change

Subjective policy uncertainty refers to the probability of a policy intervention being effective [50]. Additionally, respondents may not fully understand a proposed policy or the way it is to be implemented. Several preference studies have investigated the role of policy change in the willingness to pay for non-marketed goods [51], showing that policy uncertainty negatively influences individual valuations. Since the announcement of a pilot duty-free shopping policy for offshore tourists in Hainan, the Shenzhen government has been considering whether to follow suit, leading tourists to hesitate when shopping. Therefore, this paper examines the duty-free shopping policy to study the influence of policy changes on tourists' shopping preferences.

Despite increasing studies on tourism consumption covering a wide range of topics, there is still a gap in the knowledge in terms of the impact of uncertainty on tourism consumption preferences in the Shenzhen tourism market. Uncertainty is inherent and inevitable in tourism consumption activities, however the NCS and PC approaches fail to accurately estimate it. To bridge this knowledge gap, a new consumer preference uncertainty model is theoretically developed in this study, in which uncertainty is considered a latent variable defined by a set of influential factors. Moreover, summarizing the above literature, a conceptual framework illustrating the relationships among uncertainty (imperfect knowledge, inexperience, unfamiliarity, and policy change) and tourism consumption preferences (food, accommodation, and shopping) and corresponding influential factors is proposed (Figure 1).

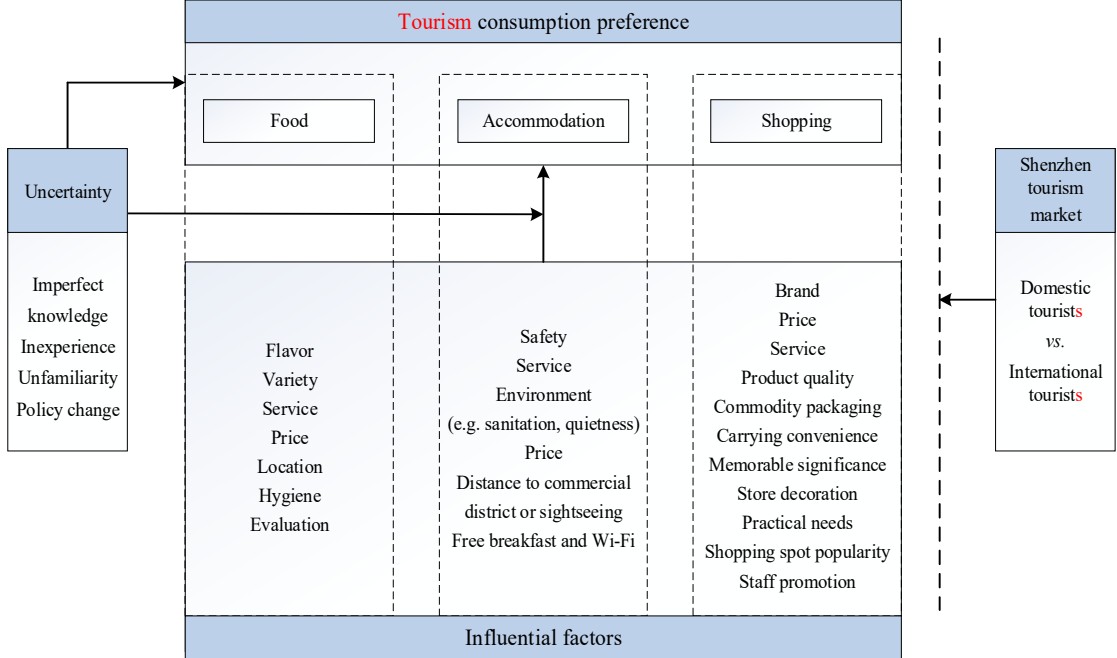

**Figure 1.** Conceptual framework.

## 3. Methodology

### 3.1. Model

The proposed model in this paper was inspired by the combination of the modified virtual expert preference approach and the ordered probit model (MVEP-OPM). The probit model is widely applied for ranked data, which decomposes the overall preferences provided by the respondents into factor-related utility weights [52]. The original virtual expert preference (OVEP) approach was first developed by [46] as a general consumer preference model for experience products and overcomes the limitations of choice models, especially when significant information on non-quantifiable attributes is missing. Due to the heavy computational load, a modified VEP (MVEP) was proposed, in which uncertainties are used instead of virtual experts to estimate the unobservable product utility. In the MVEP model, $Y_{ij}$ represents the stated preference of consumer $i$ for product $j$. $R$ denotes the number of points on the ordered scale and $r$ represents a specific response, which is modeled by postulating the existence of $R-1$ thresholds for the latent random utilities ($U_{ij}$), as below:

$$Y_{ij} = r \ if \ C_{i,r-1} < U_{ij} \leq C_{i,r} \ r = 1,2,\ldots,R$$
$$j = 1,2,3 \ indicate \ food, \ accommodation, \ shopping, \ respectively. \tag{1}$$

The latent product utility $U_{ij}$ is decomposed into the effects of certain ($U'_{ij}$) and uncertain ($U^*_{ij}$) components as:

$$U_{ij} = U'_{ij} + U^*_{ij} \tag{2}$$

Here, $U'_{ij}$ is measured by observed product attributes $X$ and individual characteristics $X^*$ as:

$$U'_{ij} = \alpha_{i0} + \alpha_{0j} + \alpha_1 X + \alpha_2 X^* + X(\Psi X^*) + \xi_{\beta i}$$
$$\xi_{\beta i} \sim N(0, \textstyle\sum_{\beta i}) \ \forall \ i \ and \ j \tag{3}$$

The intercepts $\alpha_{i0}$ and $\alpha_{0j}$ capture the main effects of consumer $i$ and product $j$, respectively; $X(\Psi X^*)$ reveals the interaction effect of consumer characteristics and product attributes; $\Psi$ is the corresponding parameter matrix, indicating the effects of observed individual characteristics on consumption preferences; $\xi_{\beta i}$ is the error term, accounting for unobserved heterogeneity across consumers.

Here, $U^*_{ij}$ is estimated by representative uncertainty influencing factors in tourism research as:

$$U^*_{ij} = \alpha_3 Z + \alpha_4 Z^2 + \varepsilon_{ij}, \ \varepsilon_{ij} \sim (0, \sigma^2) \tag{4}$$

where $Z$ is the uncertainty factor; $\alpha_3$, $\alpha_4$ are corresponding parameters. If consumer $i$ response choices are related to preference, the probability $P_{ij}$ is given by:

$$P_i(Y_{ij} = r) = \ P(C_{i,r-1} < U_{ij} \leq C_{ir})$$
$$= \Phi(C_{ir} - U_{ij}) - \Phi(C_{i,r-1} - U_{ij}) \tag{5}$$

where $\Phi(\cdot)$ is the standard normal cumulative distribution function. The log-likelihood function is:

$$\underset{(\alpha_0,\alpha_1,\alpha_2,\alpha_3,\alpha_4)}{LogL} = \sum_i \ln[P_i(Y_{ij} = r)]$$
$$= \sum_i \ln[\Phi(C_{i,r} - \alpha_0 - \alpha_1 X - \alpha_2 X^* - X \times (\Psi X^*) - \alpha_3 Z - \alpha_4 Z^2)$$
$$- \Phi(C_{i,r-1} - \alpha_0 - \alpha_1 X - \alpha_2 X^* - X \times (\Psi X^*) - \alpha_3 Z - \alpha_4 Z^2)] \tag{6}$$

The individual-specific cutoff point $C_{ir}$ accounts for scale usage heterogeneity, enabling us to cluster people based on preference similarity. The log-likelihood in Equation (6) is maximized with respect to the elements of $\alpha_0, \alpha_1, \alpha_2, \alpha_3, \alpha_4$ along with the cutoff points $C_{i1}, C_{i2}, \ldots, C_{i,r-1}$ using an iterative procedure to give maximum likelihood estimates of both sets of parameters. The definitions of variables in the model are explained in Table 1.

**Table 1.** List of variables and definitions.

| Variables | Definition | Variables | Definition |
|---|---|---|---|
| Dependent variable | tourism consumption preference | Control variable | individual characteristics |
| $Y_1$ (food) | 1 (luxury) | $X^*_1$ (gender) | female; male |
| restaurant style | 2 (ordinary) | $X^*_2$ (age) | −24; 25–34; 35–44; 45–59; 60+ |
| | 3 (local flavor) | $X^*_3$ (marital status) | single; married |
| | 4 (fast food, tea room, bar) | $X^*_4$ (income RMB/USD) | −2000; 2001–5000; 5001–10,000 |
| $Y_2$ (accommodation) | 1 (3+ star rating) | | 10,000–15,000; 15,000+ |
| hotel level | 2 (motel) | Independent variable | product attributes |
| | 3 (hostel) | PartI: Food | all independent variables |
| | 4 (friend/relative home) | $X_1$ (f_flavor) | are measured on a 5-point |
| $Y_3$ (shopping) | 1 (−10%) | $X_2$ (f_variety) | Likert scale: |
| expenditure ratio | 2 (10%–15%) | $X_3$ (f_service) | 1 (the least important) |
| | 3 (15%–20%) | $X_4$ (f_price) | 3 (neutral) |
| | 4 (20% +) | $X_5$ (f_location) | 5 (the most important) |
| Uncertainty variable | uncertainty level | $X_6$ (f_hygiene) | |
| $Z_1$ (uncertainty$_1$) | education degree | $X_7$ (f_evaluation) | |
| imperfect knowledge | 1 (middle/high school) | PartII: Accommodation | |
| | 2 (associate) | $X_8$ (a_safety) | |
| | 3 (bachelor) | $X_9$ (a_service) | |
| | 4 (master, PhD) | $X_{10}$ (a_sanitation) | |
| $Z_2$ (uncertainty$_2$) | # of trips | $X_{11}$ (a_price) | |
| inexperience | 1 (never) | $X_{12}$ (a_location) | |
| | 2 (1–2 times) | $X_{13}$ (a_free-breakfast) | |
| | 3 (3–4 times) | PartIII: Shopping | |
| | 4 (4 times+) | $X_{14}$ (s_brand) | |
| $Z_3$ (uncertainty$_3$) | duration of stay | $X_{15}$ (s_price) | |
| unfamiliarity | 1 (less than 3 days) | $X_{16}$ (s_service) | |
| | 2 (4–7days) | $X_{17}$ (s_quality) | |
| | 3 (1–2 weeks) | $X_{18}$ (s_packaging) | |
| | 4 (2 weeks or above) | $X_{19}$ (s_carrying) | |
| $Z_4$ (uncertainty$_4$) | ratio of shopping tax rebates | $X_{20}$ (s_memorable) | |
| policy change | 1 (−10%) | $X_{21}$ (s_decoartion) | |
| | 2 (10%–15%) | $X_{22}$ (s_needs) | |
| | 3 (15%–20%) | $X_{23}$ (s_popularity) | |
| | 4 (20%+) | $X_{24}$ (s_promotion) | |

*3.2. Data*

3.2.1. Data Collection

The data were obtained through questionnaires (Chinese version, CV; English version, EV), in which 35 items (3 preferences, 4 uncertainties, 4 consumer characteristics, 24 product attributes) were designed and collected by asking tourists to assess their tourism consumption preferences in terms of food (partI), accommodation (parII), and shopping (partIII) at 14 passenger gathering places in Shenzhen, including East Gate Pedestrian Street, Splendid China, Window of the World, Lotus Hill, Shenzhen Airport, Overseas Chinese Town, Coastal City, Shenzhen North Railway Station, Shenzhen Library, Shenzhen University, Peking University Shenzhen Graduate School, Yitian Holiday Plaza, Shekou Sea World, and Overseas Chinese Town Contemporary Art Terminal. In designing the questionnaire, the survey was initially presented as the Chinese version (CV) and then translated into the English version (EV). The double back-translation method, double proofreading process, and a pilot test were completed prior to data collection. In addition, at the questionnaire collection site, for each target international tourist, the questionnaire was further interpreted by two independent professional translators to avoid ambiguity caused by language barriers due to different cultural backgrounds. Respondents were free to respond to three parts, since each part is independent.

A total of 600 tourist surveys (CV 340, EV 260) were distributed and 581 surveys (CV335, 99%; EV 246, 95%) were returned. Since 46 questionnaires with multiple missing values were removed, a total of 535 valid surveys (CV 326; EV 209) were coded for data analysis, specifically for food (473), accommodation (502), and shopping (402) components.

### 3.2.2. Statistical Analysis

Demographic profile: In Table 2, according to the frequency analysis, most domestic and international tourists are men (food: 53% vs. 69%; accommodation: 53% vs. 67%; shopping: 52% vs. 68%) aged 25–34 years. However, the majority of domestic tourists visiting Shenzhen are married, with monthly personal income of RMB 2000–5000 (food, 42%; accommodation, 43%; shopping, 38%), while large numbers of international visitors are single, with income of 5000–10,000 USD/month. Specifically, for food preferences, only income ($\chi^2$ =23.36, $p$ = 0.025) has a significant impact on domestic tourists, while income ($\chi^2$ =21.54, $p$ = 0.043), marital status ($\chi^2$ =10.82, $p$ = 0.013), and age ($\chi^2$ =18.88, $p$ = 0.091) are all significant for international tourists. "Local flavor" restaurants are popular with international tourists aged over 60 or with income above 15,000 USD/month. Regarding accommodation preferences, significant effects are found on domestic tourists (DT) and international tourists (IT) with regard to gender (DT, $\chi^2$ =10.44, $p$ = 0.015 vs. IT, $\chi^2$ =12.64, $p$ = 0.005), marital status (DT, $\chi^2$ =20.11, $p$ = 0.000 vs. IT, $\chi^2$ = 9.54, $p$ = 0.023), and income (DT, $\chi^2$ = 52.53, $p$ = 0.000 vs. IT, $\chi^2$ = 33.01, $p$ = 0.001). The only difference is that the age item has a significant relationship to domestic tourists ($\chi^2$ = 38.43, $p$ = 0.000). In addition, 5000 USD/month is a watershed for distinguishing international tourists preferring luxury hotels or hostels, while domestic visitors show a preference for motels. When it comes to shopping preferences, gender ($\chi^2$ = 10.12, $p$ = 0.018) and marital status ($\chi^2$ = 22.73, $p$ = 0.000) are significant for domestic tourists, while only marital status ($\chi^2$ = 7.50, $p$ = 0.058) has a significant association with international visitors. Taking the shopping and spending rates of 15%–20% for instance, international married tourists account for 23% of the total, more than twice that of single tourists (10%).

**Table 2.** Chi-square tests used to identify associations between demographic profiles and tourism consumption preferences.

| Preference (#, %) | $X^{*}_1$ (Gender) | | $X^{*}_2$ (Age) | | | | | $X^{*}_4$ (Income, RMB or $ in Thousands) | | | | |
|---|---|---|---|---|---|---|---|---|---|---|---|---|
| | Female | Male | 15–24 s | 25–34 s | 35–44 s | 45–59s | 60 s+ | −2 | 2–5 | 5–10 | 10–15 | 15+ |
| **Chinese tourists** | | | | | | | | | | | | |
| *Food* $\chi^2$ = 6.25, $p$ = 0.100 | | | $\chi^2$ = 16.10, $p$ = 0.187 | | | | | $\chi^2$ = 23.36, $p$ = 0.025 | | | | |
| (N = 284) | n = 133(47) | n = 151(53) | n = 85(30) | n = 136(48) | n = 49(17) | n = 13(5) | n = 1(0) | n = 39(14) | n = 118(42) | n = 76(27) | n = 29(10) | n = 22(8) |
| 1 | 1(1) | 3(2) | 1(1) | 1(1) | 2(4) | 0 | 0 | 1(3) | 1(1) | 1(1) | 1(3) | 0 |
| 2 | 24(18) | 44(29) | 23(27) | 33(24) | 7(14) | 4(31) | 1(100) | 14(36) | 38(32) | 12(14) | 1(3) | 4(18) |
| 3 | 97(73) | 96(64) | 55(65) | 95(70) | 37(76) | 6(46) | 0 | 20(51) | 71(60) | 69(80) | 24(86) | 16(73) |
| 4 | 11(8) | 8(5) | 6(7) | 7(5) | 3(6) | 3(23) | 0 | 4(10) | 8(7) | 3(4) | 2(8) | 2(9) |
| *Accommodation* $\chi^2$ = 10.44, $p$ = 0.015 | | | $\chi^2$ = 38.43, $p$ = 0.000 | | | | | $\chi^2$ = 52.53, $p$ = 0.000 | | | | |
| (N = 306) | n = 143(47) | n = 163(53) | n = 91(30) | n = 149(49) | n = 53(17) | n = 12(4) | n = 1(0) | n = 39(13) | n = 133(43) | n = 84(27) | n = 27(9) | n = 23(8) |
| 1 | 28(20) | 34(21) | 11(12) | 23(15) | 22(42) | 6(50) | 0 | 7(18) | 15(11) | 16(19) | 10(37) | 14(61) |
| 2 | 76(53) | 108(66) | 53(58) | 103(69) | 24(45) | 3(25) | 1(100) | 19(49) | 82(62) | 60(71) | 16(59) | 7(30) |
| 3 | 17(12) | 8(5) | 10(11) | 9(6) | 5(9) | 1(8) | 0 | 5(13) | 14(10) | 3(4) | 1(4) | 2(9) |
| 4 | 22(15) | 13(8) | 17(19) | 14(10) | 2(4) | 2(17) | 0 | 8(20) | 22(17) | 5(6) | 0 | 0 |
| *Shopping* $\chi^2$ = 10.12, $p$ = 0.018 | | | $\chi^2$ = 17.23, $p$ = 0.141 | | | | | $\chi^2$ = 15.34, $p$ = 0.223 | | | | |
| (N = 237) | n = 113(48) | n = 124(52) | n = 69(29) | n = 125(53) | n = 35(15) | n = 7(3) | n = 1(0) | n = 32(14) | n = 91(38) | n = 72(30) | n = 23(10) | n = 19(8) |
| 1 | 30(27) | 42(34) | 19(28) | 38(30) | 13(37) | 2(29) | 0 | 11(34) | 28(31) | 22(31) | 4(17) | 7(37) |
| 2 | 52(46) | 37(30) | 35(51) | 44(35) | 8(23) | 1(13) | 1(100) | 14(44) | 34(37) | 29(40) | 7(30) | 5(26) |
| 3 | 25(22) | 27(22) | 9(13) | 29(23) | 12(34) | 2(29) | 0 | 2(6) | 23(25) | 13(18) | 10(43) | 4(21) |
| 4 | 6(5) | 18(14) | 6(8) | 14(12) | 2(6) | 2(29) | 0 | 5(16) | 6(7) | 8(11) | 2(10) | 3(16) |
| **International tourists** | | | | | | | | | | | | |
| *Food* $\chi^2$ = 0.89, $p$ = 0.829 | | | $\chi^2$ = 18.88, $p$ = 0.091 | | | | | $\chi^2$ = 21.54, $p$ = 0.043 | | | | |
| (N = 189) | n = 59(31) | n = 130(69) | n = 38(20) | n = 65(34) | n = 46(24) | n = 32(17) | n = 8(4) | n = 46(24) | n = 34(18) | n = 58(31) | n = 27(14) | n = 24(13) |
| 1 | 7(12) | 19(15) | 3(8) | 7(11) | 5(11) | 10(31) | 1(13) | 1(2) | 3(9) | 13(22) | 5(19) | 4(17) |
| 2 | 15(25) | 30(23) | 9(24) | 10(15) | 14(30) | 8(25) | 4(50) | 11(24) | 10(29) | 12(21) | 2(7) | 10(42) |
| 3 | 33(56) | 68(52) | 23(60) | 40(62) | 23(50) | 12(38) | 3(38) | 29(63) | 17(50) | 28(48) | 19(70) | 8(33) |
| 4 | 4(7) | 13(10) | 3(8) | 8(13) | 4(9) | 2(6) | 0 | 5(11) | 4(12) | 5(9) | 1(4) | 2(8) |
| *Accommodation* $\chi^2$ = 12.64, $p$ = 0.005 | | | $\chi^2$ = 12.59, $p$ = 0.399 | | | | | $\chi^2$ = 33.01, $p$ = 0.001 | | | | |
| (N = 196) | n = 64(33) | n = 132(67) | n = 43(22) | n = 61(31) | n = 48(24) | n = 35(18) | n = 9(5) | n = 51(26) | n = 35(18) | n = 57(29) | n = 28(14) | n = 25(13) |
| 1 | 17(27) | 58(44) | 13(30) | 18(30) | 21(44) | 16(46) | 7(78) | 11(22) | 8(23) | 23(40) | 15(54) | 18(72) |
| 2 | 13(20) | 22(17) | 9(21) | 11(18) | 9(19) | 5(14) | 1(11) | 11(22) | 7(20) | 11(19) | 4(14) | 2(8) |
| 3 | 28(44) | 29(22) | 14(33) | 23(38) | 11(23) | 8(23) | 1(11) | 15(29) | 14(40) | 20(35) | 5(18) | 3(12) |
| 4 | 6(9) | 23(17) | 7(16) | 9(14) | 7(14) | 6(17) | 0 | 14(27) | 6(17) | 3(6) | 4(14) | 2(8) |
| *Shopping* $\chi^2$ = 4.45, $p$ = 0.217 | | | $\chi^2$ = 17.01, $p$ = 0.149 | | | | | $\chi^2$ = 15.42, $p$ = 0.219 | | | | |
| (N = 165) | n = 52(32) | n = 113(68) | n = 39(24) | n = 60(36) | n = 38(23) | n = 21(13) | n = 7(4) | n = 44(27) | n = 36(22) | n = 43(26) | n = 23(14) | n = 19(12) |
| 1 | 17(33) | 56(50) | 19(49) | 24(40) | 15(39) | 11(52) | 4(57) | 16(36) | 16(44) | 19(44) | 8(35) | 14(74) |
| 2 | 22(42) | 39(35) | 13(33) | 18(30) | 20(53) | 9(43) | 1(14) | 19(43) | 12(33) | 14(33) | 13(57) | 3(16) |
| 3 | 11(21) | 15(13) | 5(13) | 16(27) | 2(5) | 1(5) | 2(29) | 8(18) | 7(19) | 7(16) | 2(8) | 2(10) |
| 4 | 2(4) | 3(2) | 2(5) | 2(3) | 1(13) | 0 | 0 | 1(3) | 1(4) | 3(7) | 0 | 0 |

Note: $X$ * denote control variables.

Uncertainty and preference: Table 3 demonstrates the uncertainty, preference, and comparative analyses between domestic and international visitors. (1) Food: The degree of uncertainty for this component is generally at a moderate level and "local flavor" restaurants are preferred, since the mean values of *uncertainty₁, uncertainty₂, uncertainty₃*, and *food preference* are 2.56, 2.16, 2.00, and 2.71, respectively. Influential factors such as flavor (4.14) and hygiene (4.26) are considered important. However, the analysis reveals

significant mean differences between the two groups in the uncertainty category, since the degrees of uncertainty caused by imperfect knowledge ($Mean_{DT}$ = 2.19 vs. $Mean_{IT}$ = 3.12, $t$ = −9.72, $p$ < 0.01), inexperience ($Mean_{DT}$ = 2.00 vs. $Mean_{IT}$ = 2.41, $p$ < 0.01), and unfamiliarity ($Mean_{DT}$ = 1.70 vs. $Mean_{IT}$ = 2.45, $t$ = −9.27, $p$ < 0.01) among international tourists are obviously lower than those of domestic travelers. When analyzing the differences in preferred food attributes, domestic tourists are particularly attentive to flavor ($Mean_{DT}$ = 4.28 vs. $Mean_{IT}$ = 3.92, $t$ = 3.53, $p$ < 0.01), service ($Mean_{DT}$ = 4.11 vs. $Mean_{IT}$ = 3.73, $t$ = 3.96, $p$ < 0.01), price ($Mean_{DT}$ = 3.73 vs. $Mean_{IT}$ = 3.42, $t$ = 3.27, $p$ < 0.01), and hygiene ($Mean_{DT}$ = 4.53 vs. $Mean_{IT}$ = 3.86, $t$ = 7.17, $p$ < 0.01), indicating that domestic tourists have high expectations for the local cuisine. (2) Accommodation: The degree of uncertainty caused by unfamiliarity (1.99) is higher than that caused by imperfect knowledge (2.54) and inexperience (2.16), and motels (2.15) are popular overall. Moreover, strong agreement is found between safety (4.40), sanitation (4.28), and service (4.03), however their effects are different between the two samples: safety ($Mean_{DT}$ = 4.69 vs. $Mean_{IT}$ = 3.94, $t$ = 7.97, $p$ < 0.01), service ($Mean_{DT}$ = 4.24 vs. $Mean_{IT}$ = 3.72, $t$ = 5.43, $p$ < 0.01), sanitation ($Mean_{DT}$ = 4.49 vs. $Mean_{IT}$ = 3.94, $t$ = 6.62, $p$ < 0.01), and price ($Mean_{DT}$ = 3.84 vs. $Mean_{IT}$ = 3.68, $t$ = 1.65, $p$ < 0.1). (3) Shopping: Yet again the degree of uncertainty for domestic tourism consumption preferences is higher; that is, imperfect knowledge ($Mean_{DT}$ = 2.26 vs. $Mean_{IT}$ = 3.08, $t$ = −8.01, $p$ < 0.01), inexperience ($Mean_{DT}$ = 2.01 vs. $Mean_{IT}$ = 2.41, $t$ = −4.92, $p$ < 0.01), and unfamiliarity ($Mean_{DT}$ = 1.71 vs. $Mean_{IT}$ = 2.51, $t$ = −9.08, $p$ < 0.01). In conclusion, compared with international visitors, uncertainty plays a greater role in domestic tourist shopping preferences, resulting in larger impacts of influential factors, such as brand ($Mean_{DT}$ = 3.41 vs. $Mean_{IT}$ = 3.13, $t$ = 2.27, $p$ < 0.05), quality ($Mean_{DT}$ = 4.37 vs. $Mean_{IT}$ = 4.10, $t$ = 2.87, $p$ < 0.01), and popularity ($Mean_{DT}$ = 3.43 vs. $Mean_{IT}$ = 2.98, $t$ = 3.91, $p$ < 0.01).

**Table 3.** The t test for comparative analysis between domestic and international tourists.

| Variables | Uncertainty and Preference | | | | | Mean Analysis | | |
| --- | --- | --- | --- | --- | --- | --- | --- | --- |
| | Domestic and International | | | | | Domestic vs. International | | |
| | Obs. | Mean | SD | Min | Max | Domestic | International | *t* value |
| Part I: Food | | | | | | | | |
| $Y_1$ (food) | 473 | 2.71 | 0.70 | 1 | 4 | 2.80 | 2.58 | 3.19 [b] |
| $Z_1$ (uncertainty$_1$) | 473 | 2.56 | 1.11 | 1 | 4 | 2.19 | 3.12 | −9.72 [a] |
| $Z_2$ (uncertainty$_2$) | 473 | 2.16 | 0.78 | 1 | 4 | 2.00 | 2.41 | −5.33 [a] |
| $Z_3$ (uncertainty$_3$) | 473 | 2.00 | 0.88 | 1 | 4 | 1.70 | 2.45 | −9.27 [a] |
| $X_1$ (f_flavor) | 473 | 4.14 | 1.10 | 1 | 5 | 4.28 | 3.92 | 3.53 [a] |
| $X_2$ (f_variety) | 473 | 3.64 | 1.08 | 1 | 5 | 3.67 | 3.60 | 0.69 |
| $X_3$ (f_service) | 473 | 3.96 | 1.03 | 1 | 5 | 4.11 | 3.73 | 3.96 [a] |
| $X_4$ (f_price) | 473 | 3.61 | 1.01 | 1 | 5 | 3.73 | 3.42 | 3.27 [a] |
| $X_5$ (f_location) | 473 | 3.39 | 1.10 | 1 | 5 | 3.33 | 3.48 | −1.48 |
| $X_6$ (f_hygiene) | 473 | 4.26 | 1.02 | 1 | 5 | 4.53 | 3.86 | 7.17 [a] |
| $X_7$ (f_evaluation) | 473 | 3.56 | 1.14 | 1 | 5 | 3.60 | 3.49 | 1.05 |
| Part II: Accommodation $Y_2$ (accommodation) | 502 | 2.15 | 0.96 | 1 | 4 | 2.11 | 2.20 | −1.03 |
| $Z_1$ (uncertainty$_1$) | 502 | 2.54 | 1.11 | 1 | 4 | 2.18 | 3.10 | −9.81 [a] |
| $Z_2$ (uncertainty$_2$) | 502 | 2.16 | 0.78 | 1 | 4 | 2.00 | 2.40 | −5.23 [a] |
| $Z_3$ (uncertainty$_3$) | 502 | 1.99 | 0.89 | 1 | 4 | 1.68 | 2.46 | −9.75 [a] |
| $X_8$ (a_safety) | 502 | 4.40 | 1.03 | 1 | 5 | 4.69 | 3.94 | 7.97 [a] |
| $X_9$ (a_service) | 502 | 4.03 | 1.04 | 1 | 5 | 4.24 | 3.72 | 5.43 [a] |
| $X_{10}$ (a_sanitation) | 502 | 4.28 | 0.94 | 1 | 5 | 4.49 | 3.94 | 6.62 [a] |
| $X_{11}$ (a_price) | 502 | 3.78 | 1.02 | 1 | 5 | 3.84 | 3.68 | 1.65 [c] |
| $X_{12}$(a_location) | 502 | 3.67 | 1.11 | 1 | 5 | 3.67 | 3.67 | −0.02 |
| $X_{13}$ (a_free-breakfast) | 502 | 3.29 | 1.26 | 1 | 5 | 3.26 | 3.33 | −0.61 |

**Table 3.** *Cont.*

| Variables | Uncertainty and Preference | | | | | Mean Analysis | | |
|---|---|---|---|---|---|---|---|---|
| Part III: Shopping | | | | | | | | |
| $Y_3$(shopping) | 402 | 1.98 | 0.92 | 1 | 4 | 2.12 | 1.78 | 3.73 [a] |
| $Z_1$ (uncertainty$_1$) | 402 | 2.60 | 1.09 | 1 | 4 | 2.26 | 3.08 | −8.01 [a] |
| $Z_2$ (uncertainty$_2$) | 402 | 2.17 | 0.76 | 1 | 4 | 2.01 | 2.41 | −4.92 [a] |
| $Z_3$ (uncertainty$_3$) | 402 | 2.04 | 0.90 | 1 | 4 | 1.71 | 2.51 | −9.08 [a] |
| $Z_4$ (uncertainty$_4$) | 402 | 2.40 | 1.10 | 1 | 4 | 2.46 | 2.32 | 1.29 |
| $X_{14}$ (s_brand) | 402 | 3.30 | 1.27 | 1 | 5 | 3.41 | 3.13 | 2.27 [b] |
| $X_{15}$(s_price) | 402 | 3.83 | 1.05 | 1 | 5 | 3.89 | 3.75 | 1.33 |
| $X_{16}$ (s_service) | 402 | 3.79 | 1.01 | 1 | 5 | 3.88 | 3.65 | 2.18 [b] |
| $X_{17}$ (s_quality) | 402 | 4.26 | 0.95 | 1 | 5 | 4.37 | 4.10 | 2.87 [a] |
| $X_{18}$ (s_packaging) | 402 | 3.19 | 1.11 | 1 | 5 | 3.26 | 3.10 | 1.43 |
| $X_{19}$ (s_carrying) | 402 | 3.92 | 1.07 | 1 | 5 | 4.08 | 3.70 | 3.56 [a] |
| $X_{20}$ (s_memorable) | 402 | 3.98 | 1.15 | 1 | 5 | 4.14 | 3.75 | 3.40 [a] |
| $X_{21}$ (s_decoartion) | 402 | 3.18 | 1.17 | 1 | 5 | 3.18 | 3.19 | −0.05 |
| $X_{22}$ (s_needs) | 402 | 3.94 | 1.06 | 1 | 5 | 4.16 | 3.62 | 5.25 [a] |
| $X_{23}$ (s_popularity) | 402 | 3.25 | 1.15 | 1 | 5 | 3.43 | 2.98 | 3.91 [a] |
| $X_{24}$ (s_promotion) | 402 | 2.98 | 1.19 | 1 | 5 | 3.05 | 2.87 | 1.52 |

[a] $p < 0.01$, [b] $p < 0.05$, [c] $p < 0.1$.

## 4. Results and Discussions

The Chow test results for food ($Pr > F = 0.005$), accommodation ($Pr > F = 0.068$), and shopping ($Pr > F = 0.087$) reject the null hypotheses at 1%, 5%, and 10% significance, respectively, indicating that consumption preferences differ between domestic and international visitors (Table 4).

**Table 4.** Model results for tourism consumption preferences under conditions of uncertainty.

| Variables | Food | | Accommodation | | Shopping | |
|---|---|---|---|---|---|---|
| | Domestic | International | Domestic | International | Domestic | International |
| | Model 1 | Model 2 | Model 3 | Model 4 | Model 5 | Model 6 |
| *Uncertainty* | | | | | | |
| $Z_1$ (uncertainty$_1$) | −0.5697 * (0.3290) | −0.1000 (0.0932) | −0.1895 *** (0.0674) | −0.0135 (0.0841) | −0.0474 (0.0815) | 0.1470 * (0.0877) |
| $Z_2$(uncertainty$_2$) | 0.2693 ** (0.1085) | −0.2060 * (0.1111) | −0.3162 ** (0.1230) | 0.5471 * (0.2934) | −1.0086 *** (0.3536) | 0.5384 * (0.3040) |
| $Z_3$ (uncertainty$_3$) | 0.1552 (0.5825) | −0.7917 * (0.4789) | −0.3140 *** (0.1180) | −0.8032 * (0.4680) | −0.0023 (0.1089) | −0.1436 (0.1276) |
| $Z_4$ (uncertainty$_4$) | | | | | 0.1358 ** (0.0654) | 0.8847 ** (0.4152) |
| *Sociodemographic* | | | | | | |
| $X^{*}_1$ (gender) | −0.4670 *** (0.1607) | 0.3157 * (0.1896) | −0.0503 (0.1394) | 0.0333 (0.2004) | 0.9553 * (0.5398) | 0.1540 (0.2163) |
| $X^{*}_{2,1}$ (age$_1$) | 0.5083 (0.5538) | 0.4886 (0.5047) | 0.4263 * (0.2234) | 0.8275 * (0.4293) | 0.8221 (0.8655) | 0.0821 (0.6012) |
| $X^{*}_{2,2}$ (age$_2$) | 0.2815 (0.5557) | 0.8400 ** (0.4145) | 0.4878 ** (0.2154) | 1.2239 *** (0.3742) | 0.6331 (0.8869) | 0.1082 (0.5533) |
| $X^{*}_{2,3}$ (age$_3$) | 0.1675 (0.5865) | 0.7336 * (0.4082) | 0.0557 (0.2567) | 1.2418 *** (0.3922) | 0.1609 (0.9149) | −0.4020 (0.5483) |
| $X^{*}_{2,4}$ (age$_4$) | 0.6029 (0.6768) | 0.2355 (0.4343) | 0.4147 (0.4919) | 1.3946 *** (0.4135) | 0.6388 (1.0215) | −0.4296 (0.5577) |
| $X^{*}_3$ (marital status) | −0.7613 ** (0.3224) | −0.2642 (0.3135) | −0.2697 (0.1829) | −0.3138 (0.2988) | 0.4918 ** (0.1990) | 0.6085 ** (0.2698) |
| $X^{*}_{4,1}$ (income$_1$) | −0.3464 (0.3999) | 0.1353 (0.4024) | 0.5099 (0.4144) | 1.3035 *** (0.4915) | 2.3909 ** (1.0685) | 1.0582 ** (0.4289) |
| $X^{*}_{4,2}$ (income$_2$) | −0.4224 (0.3225) | −0.0695 (0.3443) | 0.7165 * (0.3693) | 1.0435 ** (0.4839) | 2.0188 (0.8380) | 0.7823 * (0.4193) |
| $X^{*}_{4,3}$ (income$_3$) | −0.0401 (0.2909) | −0.0681 (0.3239) | 0.4524 (0.3419) | 0.7929 * (0.4800) | 1.4611 ** (0.6334) | 0.6364 (0.4007) |
| $X^{*}_{4,4}$ (income$_4$) | 0.3684 (0.3550) | −0.2978 (0.3442) | 0.1682 (0.3948) | 0.7098 (0.4861) | 1.2383 *** (0.4821) | 1.0331 ** (0.4287) |

**Table 4.** *Cont.*

| Variables | Food | | Accommodation | | Shopping | |
|---|---|---|---|---|---|---|
| | Domestic | International | Domestic | International | Domestic | International |
| | Model 1 | Model 2 | Model 3 | Model 4 | Model 5 | Model 6 |
| *Product attribute* | | | | | | |
| $X_1$ (f_flavor) | 0.1314 * (0.0781) | −0.0453 (0.0929) | | | | |
| $X_2$ (f_variety) | 0.3483 * (0.2079) | 0.1823 * (0.0966) | | | | |
| $X_3$ (f_service) | 0.1337 (0.0907) | −0.0557 (0.1095) | | | | |
| $X_4$ (f_price) | −0.0582 (0.0886) | 0.0146 (0.1067) | | | | |
| $X_5$ (f_location) | 0.0029 (0.0796) | −0.2546 ** (0.0990) | | | | |
| $X_6$ (f_hygiene) | −0.5775 * (0.2719) | −0.2452 *** (0.0872) | | | | |
| $X_7$(f_evaluation) | −0.0872 (0.0695) | −0.1925 ** (0.0889) | | | | |
| $X_8$ (a_safety) | | | 0.2429 * (0.1338) | 0.4333 ** (0.1916) | | |
| $X_9$ (a_service) | | | −0.1260 (0.1020) | −0.0673 (0.0947) | | |
| $X_{10}$(a_sanitation) | | | −0.0781 (0.1150) | −0.3441 *** (0.1201) | | |
| $X_{11}$ (a_price) | | | 0.0459 (0.0801) | 0.2726 *** (0.1010) | | |
| $X_{12}$ (a_location) | | | −0.0714 (0.0638) | −0.0579 (0.0921) | | |
| $X_{13}$ (a_free-breakfast) | | | −0.1025 * (0.0560) | 0.1385 * (0.0768) | | |
| $X_{14}$ (s_brand) | | | | | 0.1347 * (0.0757) | 0.0473 (0.1110) |
| $X_{15}$ (s_price) | | | | | −0.0629 (0.0893) | 0.4034 (0.2957) |
| $X_{16}$ (s_service) | | | | | −0.0002 (.0932) | −0.1062 (0.1240) |
| $X_{17}$ (s_quality) | | | | | 0.1113 (0.1067) | −0.0815 (0.1181) |
| $X_{18}$ (s_packaging) | | | | | −0.0758 (0.0833) | 0.3348 *** (0.1088) |
| $X_{19}$ (s_carrying) | | | | | −0.2785 *** (0.0857) | −0.0906 (0.1167) |
| $X_{20}$ (s_memorable) | | | | | −0.4297 ** (0.2227) | −0.2298 ** (0.1047) |
| $X_{21}$ (s_decoartion) | | | | | 0.0679 (0.0841) | 0.6555 ** (0.2720) |
| $X_{22}$ (s_needs) | | | | | 0.0052 (0.0896) | 0.2982 *** (0.1074) |
| $X_{23}$ (s_popularity) | | | | | 0.0089 (0.0938) | −0.0463 (0.1290) |
| $X_{24}$ (s_promotion) | | | | | 0.0971 (0.0774) | 0.1398 (0.1176) |
| *Interactive/square items* | | | | | | |
| $Z_1 * X_6$ | 0.1327 * (0.0747) | | | | | |
| $Z_3 * X_2$ | −0.1733 (0.1149) | | | | | |
| $Z_3\hat{}2$ | 0.1395 (0.1284) | 0.1486 * (0.0891) | | 0.1859 ** (0.0738) | | |
| $Z_2 * X_8$ | | | | −0.1770 ** (0.0915) | | |
| $Z_2 * X^*_4$ | | | | | 0.2936 *** (0.1067) | |
| $X^*_1 * X_{20}$ | | | | | 0.2364 ** (0.1259) | |
| $Z_2 * X_{21}$ | | | | | | −0.1788 * (0.0973) |
| $Z_4 * X_{15}$ | | | | | | −0.2465 ** (0.1047) |

**Table 4.** *Cont.*

| Variables | Food | | Accommodation | | Shopping | |
|---|---|---|---|---|---|---|
| | Domestic | International | Domestic | International | Domestic | International |
| | Model 1 | Model 2 | Model 3 | Model 4 | Model 5 | Model 6 |
| Log pseudolikelihood | −220.42 | −198.77 | −287.86 | −224.31 | −279.32 | −151.65 |
| Wald chi2(.) | (24) 61.29 | (24)48.48 | (19) 71.86 | (22)73.91 | (27) 56.83 | (27) 66.16 |
| Prob > chi2 | 0.0000 | 0.0022 | 0.0000 | 0.0000 | 0.0007 | 0.0000 |
| Chow-test | | | | | | |
| ChowF | | 2.55 | | 1.70 | | 1.54 |
| Pr > F | | 0.005 | | 0.068 | | 0.088 |

Note: \*\*\*, \*\*, \* indicate significance at 1%, 5%, and 10%, respectively.

### 4.1. Food Consumption Preference Estimation Results

Model 1 shows the results for domestic tourists' food preferences. *Uncertainty$_1$* (*Coef.* = −0.5697) and *uncertainty$_2$* (*Coef.* = 0.2693) are significant at 0.1 and 0.05 respectively, indicating that domestic tourists prefer fast food when facing a higher level of *uncertainty$_1$* (a lack of knowledge of indigenous food) and tend to seek local cuisine under lower degrees of *uncertainty$_2$* when they have experience due to an increasing number of trips, because imperfect knowledge makes it difficult for tourists to ascertain whether their expectations can be satisfied when dining in a high-class restaurant. Both *flavor* (0.1314) and *variety* (0.3483) exhibit positive effects at the 0.1 significance level, indicating that domestic tourists tend to prefer "local flavor" restaurants, confirming the statistical analysis results shown in Table 3. As local cuisine can satisfy tourists' expectations for vision, taste, and smell, enjoying local food (night market delicacies and street snacks) is a sensory pleasure event and part of a holiday experience, especially for females (*Coef.* = −0.4670, $p < 0.01$) and single (*Coef.* = −0.7613, $p < 0.05$) tourists. However, married visitors dislike fast food, night markets, and street snacks, which can be attributed to health concerns, since the hygiene coefficient is −0.5775 at the 0.1 significance level; this is particularly true for family tours with children. In addition, an interaction between *uncertainty$_1$* and *hygiene* was estimated and found to be significant (*Coef.* = 0.1327, $p < 0.1$), indicating that *uncertainty$_1$* plays a regulatory role in terms of the impacts of *hygiene* on food consumption preferences. Figure 2a shows the regulation mechanism and reveals that the lower the degree of *uncertainty$_1$*, the larger its effect on the impacts of *hygiene* on food preference.

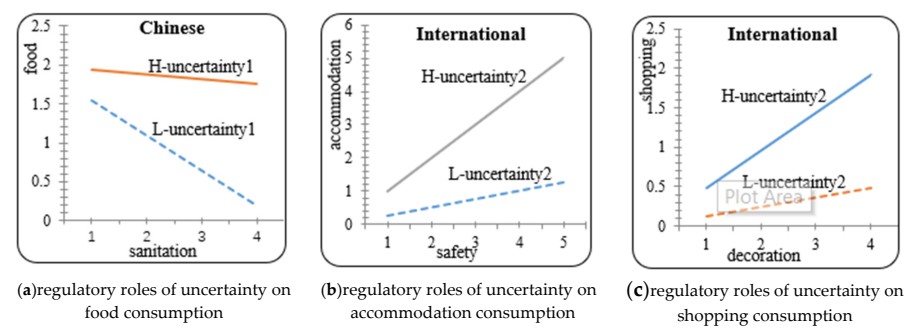

(**a**)regulatory roles of uncertainty on food consumption  (**b**)regulatory roles of uncertainty on accommodation consumption  (**c**)regulatory roles of uncertainty on shopping consumption

**Figure 2.** Regulatory roles of uncertainty on tourism consumption preferences.

Model 2 reveals the results for international tourists' food preferences. The coefficients of *uncertainty$_2$* and *uncertainty$_3$* are −0.2060 and −0.7917, respectively, which pass the 0.1 significance level test, indicating that local indigenous food loses its appeal to international tourists visiting Shenzhen with increases in the number of trips and duration of stay, differing from domestic tourists. Taking *uncertainty$_3$* as an example, as seen in Figure 3b, the second derivative of *food preference* with respect to *uncertainty$_3$* is positive and the inflection point of *uncertainty$_3$* is 2.63; that is, the duration of stay is about 11 days (see definitions in Table 2). Before the turning point, the dietary preferences of foreign tourist gradually shift from high-class restaurants to local snacks. However, when a tourist

stays at a destination for more than 11 days, their food preference once again transfers to regular restaurants rather than local flavors or fast food. Compared with domestic tourists, in addition to *variety* (*Coef.* = 0.1823, $p < 0.5$) and *sanitation* (*Coef.* = −0.2452, $p < 0.01$) discussed in model 1, international tourists are still concerned about the *location* (*Coef.* = −0.2546, $p < 0.05$) and *evaluation* (*Coef.* = −0.1925, $p < 0.05$) of restaurants, as it is inconvenient and troublesome for them to look for delicacies due to language problems. Consequently, conveniently located and highly recommended restaurants are favored by international tourists.

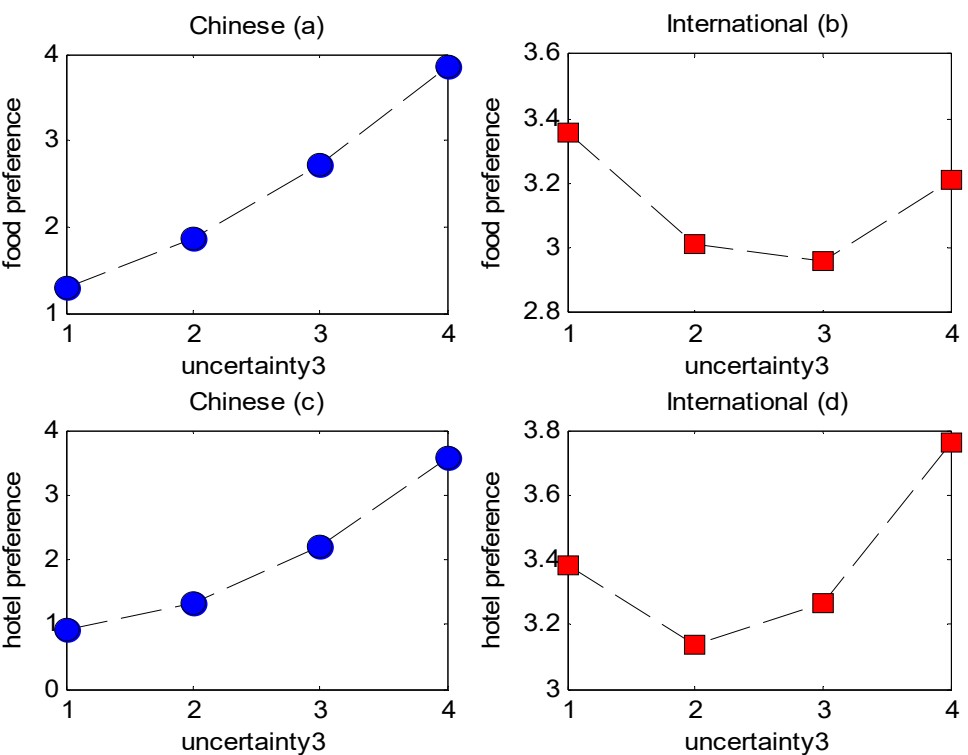

**Figure 3.** Differences in impacts of uncertainty on domestic and international tourists' preferences.

*4.2. Accommodation Consumption Preference Estimation Results*

Model 3 shows the results for domestic tourists' accommodation preferences, showing negative significant effects of *uncertainty$_1$* (*Coef.* = −0.1895, $p < 0.01$), *uncertainty$_2$* (*Coef.* = −0.3126, $p < 0.05$), and *uncertainty$_3$* (*Coef.* = −0.3140, $p < 0.01$), revealing that a high degree of uncertainty leads to a tendency of domestic tourists to lodge in the homes of friends or relatives. However, when the uncertainty is reduced—that is, through more frequent trips and a longer duration of stay—domestic tourists like to find accommodation outside (hotel or motel) rather than overnight at a friend's home, because for most families in Shenzhen an extra bedroom is not available for long-stay guests due to expensive housing prices. In addition, both *safety* (*Coef.* = 0.2429, $p < 0.1$) and *free-breakfast* (*Coef.* = −0.1025, $p < 0.1$) show significant associations with accommodation preferences, indicating that safety-conscious tourists dislike staying outside, and that the more they value free breakfasts, the greater their preference for "star" hotels.

Model 4 shows the results for international tourist accommodation preferences. *Uncertatinty$_2$* is negatively significant at the level of 0.1, indicating that international visitors favor higher class hotels under high uncertainty conditions caused by inexperience. If *uncertatinty$_3$* (*Coef.* = −0.8032, $p < 0.1$) and its square item (*Coef.* = 0.1859, $p < 0.05$) are significant, an *uncertatinty$_3$* of 2.1 would result in a minimum value for the accommodation preference, as shown in Figure 3d. In other words, for travel within 7 days, accommodation preferences gradually decline from "star" hotels to youth hostels, but accommodation options will be reversed if the stay is extended. Regarding influential

factors, *safety* (*Coef.* = 0.4333, $p < 0.05$) and its interaction with *uncertainty$_2$* (*Coef.* = $-0.1770$, $p < 0.05$) are significant, demonstrating that *uncertainty$_2$* regulates the effects of *safety* (see Figure 2b). Specifically, *safety* shows stronger and weaker impacts on accommodation preference, respectively, under high and low levels of *uncertainty$_2$*. Moreover, unlike domestic tourists, international visitors are sensitive to *sanitation* (*Coef.* = $-0.3441$, $p < 0.01$) and *price* (*Coef.* = 0.2726, $p < 0.01$) as they relate to accommodation; that is, the more they pay attention to *sanitation*, the greater their preference for superior accommodation, such as star hotels, while price-sensitive visitors show preference for cheap accommodation, such as hostels.

### 4.3. Shopping Consumption Preference Estimation Results

Model 5 provides the results for domestic tourists' shopping preferences. It shows that *uncertainty$_2$* is negatively significant at the 0.01 level, as it is well known that the more frequently a person travels to a destination, the less willing they are to shop. However, if the duty-free policy were be implemented, tourists' shopping enthusiasm would be rekindled, as shown by the estimation results for *uncertainty$_4$* (*Coef.* = 0.1358, $p < 0.05$), which show that a higher tax rebate rate would result in a larger proportion of shopping expenses. Although tourists with considerable income have the ability to pay, this process is subject to *uncertainty$_2$* due to the significant interactions (*Coef.* = 0.2936, $p < 0.01$). Popular brands with great reputations attract and stimulate spending (*Coef.* = 0.1347, $p < 0.1$), while the *price* factor is no longer a focus for domestic tourists from different financial backgrounds, as they can afford their favorite brands. Portability is becoming a key factor and items that are easy to carry increase shopping consumption (*Coef.* = $-0.2785$, $p < 0.01$). Furthermore, the *memorable* coefficient is $-0.4297$ in model 5 and passes the significance level of 5%, while its interaction with *gender* is positive (*Coef.* = 0.2364, $p < 0.05$), showing that commemorative shopping preferences will undermine consumer spending, especially for female tourists.

Model 6 covers the results for international tourists' shopping preferences. Significance is found for *uncertainty$_1$* caused by imperfect knowledge (*Coef.* = 0.1470, $p < 0.1$), *uncertainty$_2$* arising from inexperience (*Coef.* = 0.5384, $p < 0.1$), and *uncertainty$_3$* relating to policy change (*Coef.* = 0.8847, $p < 0.05$). For international visitors, *packing* (*Coef.* = 0.3348, $p < 0.01$), *needs* (*Coef.* = 0.2982, $p < 0.01$), and *decoration* (*Coef.* = 0.6555, $p < 0.05$) are key factors influencing their shopping preferences, differing from domestic tourists. Regarding the *memorable* (*Coef.* = $-0.2298$, $p < 0.05$) component, the conclusion is the same as discussed in model 5. In addition, the moderating role of *uncertainty$_2$* on *decoration* is shown in Figure 2c, whereby high uncertainty$_2$ significantly increases the impacts of in-store decorations on international tourists' shopping expenses.

### 4.4. Marginal Utilities

The marginal utility of preference is an important indicator that measures the sensitivity of preference in response to uncertainty and influential factors. For food and accommodation, the results are shown in Tables 5 and 6. The marginal utility of preferences is discussed only by taking shopping consumption as an example (Table 7). Considering that the value of shopping preference is 1, increasing *uncertainty$_2$* by 1% would lead to a 31.16% increase in domestic tourist shopping expenditure, while also resulting in a 15.92% decrease for international visitors. Moreover, with a 1% increase in *uncertainty$_4$*, the shopping expenditure of domestic and international tourists will likely decrease by 4.19% and 26.16%, respectively. Domestic and international visitors also exhibit different levels of sensitivity to the changes of influential factors, such as *brand, packaging, carrying, memorable, decoration,* and *needs* components, due to their significant marginal utilities. For example, domestic tourists are highly sensitive to the *brand* ($-0.0416$) component in the model, while the practical *needs* ($-0.0990$) component is more flexible in terms of expenditure for international visitors, further confirming that domestic tourists regard shopping as a necessity.

**Table 5.** The marginal utilities of food preferences.

| Variable | Food | | | | | | | |
|---|---|---|---|---|---|---|---|---|
| | $Y_1 = 1$ | | $Y_1 = 2$ | | $Y_1 = 3$ | | $Y_1 = 4$ | |
| | D | I | D | I | D | I | D | I |
| $Z_1$ | 0.0194 | n.a. | 0.1450 * | n.a. | −0.0967 * | n.a. | −0.0677 | n.a. |
| | (1.42) | n.a. | (1.76) | n.a. | (−1.75) | n.a. | (−1.64) | n.a. |
| $Z_2$ | −0.0092 * | 0.0391 * | −0.0685 ** | 0.0291 * | 0.0457 ** | −0.0386 * | 0.0320 * | −0.0296 * |
| | (−1.75) | (1.84) | (−2.46) | (1.84) | (2.23) | (−1.88) | (2.47) | (−1.74) |
| $Z_3$ | n.a. | 0.1503 * | n.a. | 0.1119 * | n.a. | −0.1483 * | n.a. | −0.1139 |
| | n.a. | (1.65) | n.a. | (1.66) | n.a. | (−1.66) | n.a. | (−1.6) |
| $X_1$ | −0.0045 | n.a. | −0.0334 * | n.a. | 0.0223 * | n.a. | 0.0156 | n.a. |
| | (−1.52) | n.a. | (−1.69) | n.a. | (1.68) | n.a. | (1.64) | n.a. |
| $X_2$ | −0.0119 | −0.0346 * | −0.0886 * | −0.0258 ** | 0.0591 * | 0.0342 * | 0.0414 | 0.0262 * |
| | (−1.31) | (−1.78) | (−1.71) | (−1.98) | (1.75) | (1.83) | (1.52) | (1.83) |
| $X_5$ | n.a. | 0.0483 *** | n.a. | 0.0360 ** | n.a. | −0.0477 ** | n.a. | −0.0366 ** |
| | n.a. | (2.56) | n.a. | (2.44) | n.a. | (−2.41) | n.a. | (−2.51) |
| $X_6$ | 0.0197 | −0.0466 *** | 0.1470 ** | −0.0347 *** | −0.0980 ** | 0.0459 *** | −0.0686 ** | 0.0353 ** |
| | (1.62) | (−2.59) | (2.16) | (−2.84) | (−2.1) | (2.68) | (−2) | (2.54) |
| $X_7$ | n.a. | 0.0366 ** | n.a. | 0.0272 ** | n.a. | −0.0361 ** | n.a. | −0.0277 ** |
| | n.a. | (2.05) | n.a. | (2.21) | n.a. | (−2.06) | n.a. | (−2.09) |

Note: n.a. indicates non-significant marginal effect; ***, **, * indicate significance at the 1%, 5%, and 10%, respectively; D and I represent domestic and international tourists.

**Table 6.** The marginal utilities of accommodation preferences.

| Variable | Accommodation | | | | | | | |
|---|---|---|---|---|---|---|---|---|
| | $Y_2 = 1$ | | $Y_2 = 2$ | | $Y_2 = 3$ | | $Y_2 = 4$ | |
| | D | I | D | I | D | I | D | I |
| $Z_1$ | 0.0436 *** | n.a. | 0.0014 | n.a. | −0.0135 *** | n.a. | −0.0316 *** | n.a. |
| | (2.82) | n.a. | (0.34) | n.a. | (−2.6) | n.a. | (−2.69) | n.a. |
| $Z_2$ | 0.0727 *** | −0.1654 * | 0.0024 | −0.0086 | −0.0225 ** | 0.0673 * | −0.0527 ** | 0.1067 * |
| | (2.63) | (−1.88) | (0.33) | (−1.16) | (−2.42) | (1.79) | (−2.44) | (1.86) |
| $Z_3$ | 0.0722 *** | 0.2428 * | 0.0024 | 0.0126 | −0.0223 ** | −0.0988 Tasble | −0.0523 *** | −0.1567 * |
| | (2.64) | (1.74) | (0.34) | (1.12) | (−2.54) | (−1.74) | (−2.62) | (−1.68) |
| $X_8$ | −0.0559 * | −0.1310 ** | −0.0019 | −0.0068 | 0.0173 * | 0.0533 ** | 0.0405 * | 0.0845 ** |
| | (−1.81) | (−2.28) | (−0.34) | (−1.26) | (1.75) | (2.08) | (1.83) | (2.31) |
| $X_{10}$ | n.a. | 0.1040 *** | n.a. | 0.0054 | n.a. | −0.0423 *** | n.a. | −0.0671 *** |
| | n.a. | (3.02) | n.a. | (1.28) | n.a. | (−2.81) | n.a. | (−2.85) |
| $X_{11}$ | n.a. | −0.0824 *** | n.a. | −0.0043 | n.a. | 0.0335 *** | n.a. | 0.0532 *** |
| | n.a. | (−2.83) | n.a. | (−1.27) | n.a. | (2.59) | n.a. | (2.74) |
| $X_{13}$ | 0.0236 * | −0.0419 * | 0.0008 | −0.0022 | −0.0073 * | 0.0170 * | −0.0171 * | 0.0270 * |
| | (1.83) | (−1.83) | (0.33) | (−1.06) | (−1.68) | (1.82) | (−1.8) | (1.75) |

Note: n.a. indicates non-significant marginal effect; ***, **, * indicate significance at the 1%, 5%, and 10%, respectively; D and I represent domestic and international tourists.

**Table 7.** The marginal utilities of shopping preferences.

| Variable | Shopping | | | | | | | |
|---|---|---|---|---|---|---|---|---|
| | $Y_3 = 1$ | | $Y_3 = 2$ | | $Y_3 = 3$ | | $Y_3 = 4$ | |
| | D | I | D | I | D | I | C | I |
| $Z_1$ | n.a. | −0.0435 * | n.a. | 0.0129 * | n.a. | 0.0222 | n.a. | 0.0084 |
| | n.a. | (−1.70) | n.a. | (1.66) | n.a. | (1.60) | n.a. | (1.54) |
| $Z_2$ | .3116 *** | −0.1592 * | 0.0022 | 0.0472 | −0.1641 *** | 0.0812 * | −0.1497 *** | 0.0308 |
| | (2.94) | (−1.78) | (0.11) | (1.63) | (−2.85) | (1.77) | (−2.72) | (1.52) |
| $Z_4$ | −0.0419 ** | −0.2616 ** | −0.0003 | 0.0776 ** | 0.0221 ** | 0.1334 ** | 0.0202 * | 0.0505 * |
| | (−2.12) | (−2.19) | (−0.11) | (2.00) | (2.12) | (2.16) | (1.93) | (1.69) |
| $X_{14}$ | −0.0416 * | n.a. | −0.0003 | n.a. | 0.0219 * | n.a. | 0.0200 * | n.a. |
| | (−1.82) | n.a. | (−0.11) | n.a. | (1.79) | n.a. | (1.73) | n.a. |
| $X_{18}$ | n.a. | −0.0990 *** | n.a. | 0.0294 *** | n.a. | 0.0505 *** | n.a. | 0.0191 ** |
| | n.a. | (−3.18) | n.a. | (2.63) | n.a. | (2.95) | n.a. | (2.22) |
| $X_{19}$ | 0.0860 *** | n.a. | 0.0006 | n.a. | −0.0453 *** | n.a. | −0.0413 *** | n.a. |
| | (3.29) | n.a. | (0.11) | n.a. | (−3.22) | n.a. | (−2.98) | n.a. |
| $X_{20}$ | 0.1327 ** | 0.0679 ** | 0.0009 | −0.0202 * | −0.0699 * | −0.0346 ** | −0.0638 * | −0.0131 ** |
| | (1.96) | (2.25) | (0.11) | (−1.92) | (−1.90) | (−2.12) | (−1.93) | (−2.04) |
| $X_{21}$ | n.a. | −0.1938 ** | n.a. | 0.0575 ** | n.a. | 0.0988 ** | n.a. | 0.0374 * |
| | n.a. | (−2.48) | n.a. | (2.17) | n.a. | (2.45) | n.a. | (1.84) |
| $X_{22}$ | n.a. | −0.0882 *** | n.a. | 0.0262 *** | n.a. | 0.0450 *** | n.a. | 0.0170 * |
| | n.a. | (−2.93) | n.a. | (2.97) | n.a. | (2.65) | n.a. | (1.91) |

Note: n.a. indicates non-significant marginal effect; ***, **, * indicate significance at the 1%, 5%, and 10%, respectively; D and I represent domestic and international tourists.

## 5. Conclusions and Implications

### 5.1. Conclusions

In this study, we applied a new proposed MVEP-OPM model to explore the tourism consumption preferences under uncertainty conditions in Shenzhen. In the MVEP-OPM, the virtual expert preference approach was replaced by uncertainty to efficiently research the unobserved components of certain utilities, whereby uncertainty was considered a latent variable and measured using a set of influential factors. In addition, three key tourism activities, namely food, accommodation, and shopping, were examined using sample responses from domestic and international tourists, providing insights into the tourism consumption preferences in response to changes in uncertainty and influential factors.

Some of significant findings were the overall higher degree of uncertainty caused by imperfect knowledge, inexperience, unfamiliarity, and policy changes; the greater impacts of uncertainty on tourism consumption preferences; and the more obvious regulating effects on other influential factors. The empirical results also revealed different types of tourism consumption preferences in models 1 to 6. Regarding food preferences, generally both domestic and international tourists preferred "local flavor" restaurants; however, domestic tourists thought highly of flavor, while international tourists paid more attention to recommendations and location. Moreover, the lower the degree of uncertainty, the larger its regulating effect on food hygiene preferences (see Figure 2a). For accommodation preferences, sanitation and price were deemed as important influential factors by international visitors, but they were not significant for domestic tourists. With reference to shopping preferences, domestic tourists regarded shopping activities in Shenzhen as a necessity, while international tourists preferred accommodation experiences and their expenditure elasticity for shopping was much higher than for other product categories. Furthermore, domestic tourists were interested in famous brands and merchandise that was easy to carry, while international tourists were more concerned by packaging and in-store decorations. The moderating role of uncertainty on decoration was regulated by the degree of uncertainty caused by inexperience (see Figure 3c). Finally, regarding the differences observed in tourism consumption preferences above, these can be largely attributed to cultural background, income, marital status, gender, and the degree of uncertainty.

*5.2. Implications*

Understanding the various types of tourism consumption preferences under uncertainty conditions is of great significance to the marketing strategy for Shenzhen, such as in market segmentation and the implementation of a duty-free policy. Based on the above empirical results and discussions, the following recommendations are made:

For food consumption, since both domestic and foreign tourists attached great importance to food hygiene and variety, the Shenzhen Food Safety Bureau, as the main supervision department, should actively play a guiding role, focusing on improving food safety and hygiene in the catering industry in tourism areas, for example by strengthening the governance, implementing dynamic supervision, and ensuring the disclosure of food safety information. The main reason for visitors losing confidence in local food consumption is that in the era of rapid socioeconomic development, the population flow has led to the spread and homogenization of food, resulting in the food being similar in different regions. Therefore, regarding food operators in scenic spots or business districts, in addition to focusing on sanitation conditions and the dining environment, food diversity should be improved in order to improve the dining experience and relieve taste fatigue. In addition, due to uncertainty caused by imperfect knowledge, inexperience, and unfamiliarity, the food consumption preferences of domestic and international tourists are quite different. Domestic tourists focus on flavor, while international visitors focus on the location and recommendations. According to the experimental results, the more importance tourists place on flavor and variety, the greater their preference for local restaurants and street snacks. However, greater consideration of location and recommendations will lead to larger opportunities for high-class restaurants. Therefore, local restaurants and street snacks are suggested for domestic tourists as key consumer groups, while senior restaurant chains with convenient locations and that come highly recommended should be prepared to better serve international tourists. Meanwhile, to minimize the degree of uncertainty arising from imperfect knowledge, inexperience, and unfamiliarity, it is recommended that the Shenzhen Tourist Administration strengthen the accessibility for international tourists, such as by advertising restaurant logos in tourist areas, promoting official websites, and providing English labeling at restaurants;

Travel accommodation is usually accompanied by certain population movements, so safety was deemed important by both groups. To prevent fraud, theft, and cheating, it is advised that security departments strengthen their supervision over the standardization of hotels near tourist areas, and that hotel managers regulate the management of information related to tourists. In addition, based on empirical results, if international tourists travel to Shenzhen with a high degree of uncertainty caused by unfamiliarity, their accommodation preference tends to be for star hotels. Specifically, their hotel preference will gradually drop from star hotels to hostels within a 7-day period. However, with the extension of this stay, the situation reverses. Therefore, hostels in Shenzhen should attract international tourists for 7-day stays, while star hotels should advertise to short-term tourists (less than 3 days) or long-term tourists (more than two weeks);

For shopping consumption, facing fierce competition in Guangdong-Hong Kong-Macao Greater Bay Area, Shenzhen's tourism industry must enhance its competitive advantages to prevent the outflow of consumption potential. The empirical results showed that a lower degree of uncertainty arising from policy changes would contribute to a greater proportion of shopping spending, providing guidance for public agencies in Shenzhen in formulating and evaluating the effectiveness of tourism policies. Therefore, it is necessary to implement a duty-free policy in Shenzhen and further improve the Qianhai–Shenzhen–Hong Kong Modern Service Industry Zone. For the majority of domestic tourists, Hong Kong is their main shopping detination. Since the multiple entry permit policy was established in 2009, tourists and Shenzhen citizens have enjoyed convenient travel and shopping in Hong Kong, which has affected Shenzhen's tourism industry to some extent. The implementation of a duty-free policy could ensure that quality imported goods are available at competitive prices, meaning that tourists do not have to travel to Hong Kong to access

global products. In addition, retail stores in Shenzhen should consider innovative marketing, such as boosting brand awareness, focusing on product packaging, making items easy to carry, and improving in-store environments. However, price cuts are unlikely to be effective, as this study showed that most tourists perceive retail products as price-inelastic.

Despite applying a new method to measure uncertainty and developing a model to examine the relationships between tourism consumption preferences and uncertainty, there are still some limitations. (1) Due to indicator restrictions in measuring uncertainty, this study only examined tourism consumption preferences under uncertainty conditions related to imperfect knowledge, inexperience, unfamiliarity, and policy changes. In fact, as mentioned in the literature review, age, income, war and political instability, health, crime, and terrorism are also factors causing uncertainty. In future research, these various uncertainty factors could be investigated as they relate to tourism consumption activities, such as transportation and entertainment. (2) For the questionnaire survey used for international tourists, although the double back-translation method, double proofreading process, and a pilot test were adopted, these measures only ensured the rigor and standardization of the process, but did not mitigate situations in which both the investigator and respondent are not native speakers of English, such as Japanese and Korean tourists.

In addition, it needs to be emphasized that one of the main challenges for Shenzhen's tourism industry is how to prepare for the upcoming generational shift in order to pursue sustainable development and transformation. In fact, a large number of well-known tourist attractions in Shenzhen, such as Splendid China and Windows of the World, were built in the 1990s, carrying the dreams of the previous generation during that special era in the early stage of China's reform and opening up. However, these attractions do not interest the new generation of young people. In the future, it would be worth verifying of proposed MVEP-OPM model from the perspective of transgenerational analysis [53]. Regarding the response to the coronavirus disease 2019 (COVID-19) pandemic and the regeneration of post-COVID-19 urban economies, the scope of the models applications will be expanded in conjunction with urban tourism planning.

**Author Contributions:** Conceptualization, X.L. and J.G.; methodology, X.L. and T.R.; software, X.L.; validation, X.L., J.G., and T.R.; formal analysis, X.L. and J.G.; investigation, X.L.; resources, X.L.; data curation, X.L.; writing—original draft preparation, X.L.; writing—review and editing, X.L., J.G., and T.R.; visualization, X.L.; supervision, J.G. and T.R.; project administration, J.G.; funding acquisition, J.G. All authors have read and agreed to the published version of the manuscript.

**Funding:** This work was supported by the Key Project of National Natural Science Foundation of China under rant no. 71831005 and the National Natural Science Foundation of China under grant no. 71402039.

**Informed Consent Statement:** It is reviewed and informed by Harbin Institute of Technology Shenzhen.

**Data Availability Statement:** The data were obtained through questionnaires (Chinese version, CV; English version, EV), in which 35 items (3 preferences, 4 uncertainties, 4 consumer characteristics, 24 product attributes) were designed and measured by asking tourists to assess their tourism consumption preferences in terms of food (partI), accommodation (partII), and shopping (partIII) at 14 passenger gathering places in Shenzhen, including East Gate Pedestrian Street, Splendid China, Window of The World, Lotus Hill, Shenzhen Airport, Overseas Chinese Town, Coastal City, Shenzhen North Railway Station, Shenzhen Library, Shenzhen University, Peking University Shenzhen Graduate School, Yitian Holiday Plaza, Shekou Sea World, and OCT Contemporary Art Terminal. The double translation method and a pilot test were completed prior to data collection, and respondents were free to respond to the three parts, since each part was independent.

**Conflicts of Interest:** The authors declare no conflict of interest.

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
