# Peer review of "Uncertainty and Tourism Consumption Preferences: Evidence from the Representative Chinese City of Shenzhen"

_sustainability, doi:10.3390/su13084103_

Round 1

Reviewer 1 Report

Overall: An interesting approach to a fundamental issue in tourism consumption.  

The authors are encouraged to rename the paper.  Sustainable Development and transformation are not the focus of the paper.

Relationship to Literature | Authors are encouraged to develop ideas that are relevant ( or remove material that is not directly relevant). Foe example on p3line 104 there is a throwaway sentence about destination image that is followed by general economic data.

Methodology: The researchers treat “international visitors” (English speaking)  as a homogeneous group. They are encouraged to address their approach to international visitors and to recognize the limitations of this approach.

Results/Conclusions/implications: the results are well presented. The conclusions and implications are reasonable.

Author Response

Point 1: An interesting approach to a fundamental issue in tourism consumption

Response 1: The authors greatly appreciate your efforts in improving the paper. The detailed responses and how we revised each comment are enclosed as follows.

Point 2: The authors are encouraged to rename the paper. Sustainable Development and transformation are not the focus of the paper.

Response 2: According to your enlightening comment, the title has been adjusted and revised to “Uncertainty and Tourism Consumption Preference: Evidence from a Chinese Representative City of Shenzhen”. The renamed title focuses on uncertainties and preferences, which is more conducive to highlighting the research content of the subject. Because uncertainties and preferences are core explanatory variables and dependent variables of this article, respectively, and should be given full attention. As for “sustainable development and transformation”, it is closely related to the research background and policy recommendations, which is indeed not suitable to be reflected in the prominent position in the topic. For corresponding changes to the title, please see lines 2-3 on page 1

Point 3: Relationship to Literature | Authors are encouraged to develop ideas that are relevant (or remove material that is not directly relevant). Foe example on p3line 104 there is a throwaway sentence about destination image that is followed by general economic data.

Response 3: Thanks for your constructive comments in revising the manuscript. We carefully and systematically reviewed the literature and deleted unnecessary and irrelevant materials. Please note that the specific revisions are clearly highlighted, using the "Track Changes" function in Microsoft Word (see p.3 line 105, 107-108).

Point 4: Methodology: The researchers treat “international visitors” (English speaking) as a homogeneous group. They are encouraged to address their approach to international visitors and to recognize the limitations of this approach.

Response 4: Regarding this comment, two changes have been made.

The first modification is reflected in the “3.3.1 data collection” section(see p.3-8 line 269-274), as follows:

In designing the questionnaire, the survey was initially written in Chinese Version (CV) and then translated into English Version (EV). The double-back-translation method, dou-ble-proofreading process, as well as a pilot test were completed prior to data collection. In addition, at the questionnaire collection site, for each target international tourist, the questionnaire will be further interpreted by two independent professional translators to avoid ambiguity caused by language barriers in different cultural backgrounds.

The second adjustment is reflected in limitations (see p.19 line 556-560), as follows:

For the questionnaire survey of international tourists, although the double-back-translation method, double-proofreading process, and a pilot test were adopted, this only ensured the rigor and standardization of the process, and did not solve the special situation that both the investigator and the respondent are not native speakers of English, such as Japanese and Korean tourists.

Point 5: Results/Conclusions/implications: the results are well presented. The conclusions and implications are reasonable

Response 5: Thank you for the comments on our paper, we found the reviews of our manuscript insightful and helpful in improving the paper. We extend our great thanks to the reviewer.

Reviewer 2 Report

Dear Authors,

Your article is interesting as well as well written and worth publishing. The proposed MVEP-OPM model seems consistent and logical. In the future, it is worth considering its verification and analysis in the perspective between generations (e.g. Kowalczyk-AnioÅ‚ & Nowacki (2018). Tourist consumption of a contemporary city-a transgenerational analysis). One can also consider its use in urban tourism planning in the phase of response to Covid-19 and the regeneration of the post-Covid -19 urban economy. A small note - perhaps it would be better to use the plural in Figure 1 (in the field: Shenzhen tourism market) and write "Domestic tourists" instead of "Domestic tourist". Then "International tourists" instead of "International tourist". Please also check whether the use of the two terms "tourist consumption" and "tourism consumption" is intentional or just an oversight. 

Author Response

Response: Thanks for your constructive comments in revising the manuscript. We have further improved the paper quality based on your suggestions. Please note that your comment is subdivided into two sub-comments, and detailed responses are given respectively.

Point 1: Your article is interesting as well as well written and worth publishing. The proposed MVEP-OPM model seems consistent and logical. In the future, it is worth considering its verification and analysis in the perspective between generations (e.g. Kowalczyk-Anioł & Nowacki (2018). Tourist consumption of a contemporary city-a transgenerational analysis). One can also consider its use in urban tourism planning in the phase of response to Covid-19 and the regeneration of the post-Covid -19 urban economy.

Response 1: We sincerely appreciate your enlightening comments. The perspective of transgenerational analysis is very useful for the prospects of the follow-up research, and is conducive to the urban tourism planning of Shenzhen during the COVID-19 epidemic. In view of this, the new relevant content is as follows(see p.19-20 line 561-570):

*********************************************************************************

In addition, it needs to be emphasized that, as one of the most challenges of Shenzhen’s tourism industry, how to prepare for the generational shift to pursue sustainable development and transformation is imminent. In fact, a large number of well-known tourist attractions in Shenzhen, such as Splendid China and Windows of the World, were built around the 1990s, carrying the dreams of the previous generation during that special era in the early stage of China’s reform and opening up. However, it is not attractive to the new generation of young people. In the future, it is worth considering the verification of the proposed mVEP-OPM model from the perspective of transgenerational analysis[53]. Especially in the phase of response to Corona Virus Disease 2019(Covid-19) and the regeneration of the post-Covid -19 urban economy, to expand its scope of application, the scope of its application will be expanded in conjunction with urban tourism planning.

Reference:

  1. Kowalczyk-Anio, J.; Nowacki, M. J. E. P. T., Tourist consumption of a contemporary city - a transgenerational analysis. 2018, 44, 7-15

Point 2: A small note - perhaps it would be better to use the plural in Figure 1 (in the field: Shenzhen tourism market) and write "Domestic tourists" instead of "Domestic tourist". Then "International tourists" instead of "International tourist". Please also check whether the use of the two terms "tourist consumption" and "tourism consumption" is intentional or just an oversight.

Response 2: The authors greatly appreciate your efforts in improving the paper. First of all, according to the suggestion, we replaced the singular form of “tourist” with the plural. Specifically, “Domestic tourists” and “International tourists” are used instead of “Domestic tourist” and “International tourist”, respectively, as shown in Figure 1(see p.5 line 221). Secondly, in order to avoid ambiguity, the term “tourist consumption” has been checked throughout and adjusted to “tourism consumption”(see p.2 line 76, 85; p.3 line 126; p.4 line 185; p.7 line 258, 263; p.9 line 325; p.18 line 461,467; p.19 line 550,552).
